# A Natural Astragalus-Based Nutritional Supplement Lengthens Telomeres in a Middle-Aged Population: A Randomized, Double-Blind, Placebo-Controlled Study

**DOI:** 10.3390/nu16172963

**Published:** 2024-09-03

**Authors:** Christophe de Jaeger, Saskia Kruiskamp, Elena Voronska, Carla Lamberti, Hani Baramki, Jean Louis Beaudeux, Patrick Cherin

**Affiliations:** 1Institute of Medicine and Physiology of Longevity (Institut de Jaeger), 127, rue de la Faisanderie, 75016 Paris, France; saskia.kruiskamp@institut-prevention-sante-longevite.fr (S.K.); elena.voronska@institut-prevention-sante-longevite.fr (E.V.); carla.lamberti1@gmail.com (C.L.); hbaramki@gmail.com (H.B.); 2Service de Biochimie Générale, Université Paris Cité, Assistance Publique-Hôpitaux de Paris, CHU Necker, Inserm UMR S_1139, 75015 Paris, France; jean-louis.beaudeux@u-paris.fr; 3Service de Médecine Interne 2, Institut E3M, Sorbonne Université, Assistance Publique-Hôpitaux de Paris, CHU Pitié-Salpêtrière, French National, Referral Center for Autoimmune Disorders, Inserm UMRS, Centre d’Immunologie et des Maladies Infectieuses (CIMI-Paris), 75013 Paris, France; patrick.cherin@wanadoo.fr

**Keywords:** Astragalus, telomerase activator, telomere length, aging, cardiovascular health, randomized controlled trial

## Abstract

Telomeres are ribonucleoprotein structures that form a protective buffer at the ends of chromosomes, maintaining genomic integrity during the cell cycle. A decrease in average telomere length is associated with with age and with aging-related diseases such as cancer and cardiovascular disease. In this study, we conducted a randomized, double-blind, placebo-controlled trial over six months to compare the effects of the Astragalus-based supplement versus a placebo on telomere length (TL) in 40 healthy volunteers (mean age 56.1 ± 6.0 years). Twenty subjects received the supplement, and 20 received placebo capsules. All participants completed the study, and no adverse side effects were reported at six months. Subjects taking the Astragalus-based supplement exhibited significantly longer median TL (*p* = 0.01) and short TL (*p* = 0.004), along with a lower percentage of short telomeres, over the six-month period, while the placebo group showed no change in TL. This trial confirmed that the supplement significantly lengthens both median and short telomeres by increasing telomerase activity and reducing the percentage of short telomeres (<3 Kbp) in a statistically and possibly clinically significant manner. These results align with a previous open prospective trial, which found no toxicity associated with the supplement’s intake. These findings suggest that this Astragalus-based supplement warrants further investigation for its potential benefits in promoting health, extending life expectancy, and supporting healthy aging.

## 1. Introduction

Astragalus, a plant widely used in traditional Chinese medicine, has garnered significant attention for its potential to activate telomerase and extend telomere length, making it a promising natural nutritional supplement for promoting healthy aging. Telomeres are ribonucleoprotein structures that form a protective buffer at the ends of chromosomes, thus maintaining genomic integrity during the cell cycle [1]. These structures consist of tandem repeats of the nucleotide sequence TTAGGG, associated with various regulatory proteins, including telomerase, the only enzyme capable of replicating telomeres [2,3].

In the absence of telomerase activity, which depends on the catalytic subunit telomerase reverse transcriptase (TERT), the ends of DNA are shortened by approximately 50 to 200 base pairs during each S phase of the cell cycle [4]. Cells that reach a critically low telomere length (TL) can no longer divide and thus undergo senescence or apoptosis [5,6,7]. Telomeres play a crucial role in preserving genome integrity by preventing chromosome ends from being recognized as DNA damage.

During each cell replication cycle, genetic material is lost, but since telomeres do not contain coding sequences, there is no loss of genomic information. Without an effective telomere maintenance mechanism, cell division will ultimately lead to the formation of short telomeres. These short telomeres lose their protective function and are reported to the cell as damaged DNA, activating cellular senescence pathways such as p53 and pRb/p16 [8], which interrupt cellular proliferation and induce senescence or apoptosis, depending on the cell type involved.

The Hayflick limit, the maximum number of divisions a cell can undergo [4,6], establishes a link between telomere length and cell lifespan. A decrease in TL, a marker of cellular aging, is associated with age and with aging-related diseases [1,5,9]. The rate of telomere shortening is influenced by environmental factors, including diet, physical activity, and lifestyle choices [10,11,12].

Astragalus contains active compounds such as astragaloside IV and cycloastragenol, which have been identified as potent telomerase activators. These compounds can compensate for replicative telomere erosion by activating telomerase, a specialized reverse transcriptase that uses a specific template RNA to extend the 3′ strand of chromosome ends. There are also telomerase-independent telomere-lengthening mechanisms based on homologous recombination events, known as ALT (alternative lengthening of telomeres) [13].

The average size of leukocyte telomeres at a given age results from three variables: inherited length, the rate of immune cell proliferation, and exposure to chronic oxidative stress. Chronic oxidative stress has been shown to be a major causal factor in telomere shortening and cellular senescence [14]. Average telomere size, which can be measured in peripheral blood leukocytes using various techniques, serves as a marker of biological age and chronic stress exposure under different physiological and pathological conditions [15]. In humans, telomere size decreases from about 10 kbp at birth to 4 kbp at 80 years of age, with a coding rate of a few dozen bases per year. The length of the telomeric sequence, which is shorter in men than in women, is linearly and inversely correlated with age and is largely genetically determined (70–80% heritability), exhibiting broad variability.

A large longitudinal population-based cohort study on subjects aged 50 and older demonstrated that while telomere length declines with age, telomere size may vary over time [16]. The study found that TL shortened in 66.32% of the cohort, remained stable in 11.23%, and lengthened in 22.45% [16]. Women of the same age showed a lower within-individual leukocyte TL shortening rate than men [16].

Numerous studies have shown that telomere shortening in peripheral blood leukocytes is a risk factor for cardiovascular disease (atherosclerosis, early infarction, hypertension, vascular dementia), metabolic disorders (diabetes, obesity, insulin resistance), mental pathologies, infections, and cancer [15,17,18,19,20,21,22,23,24,25,26]. Mortality from infections or cardiovascular disease is three to eight times higher in individuals over 60 years of age, with the shortest telomeres compared to those with the longest telomeres [27]. This underscores the pivotal role of telomeres at the interface of molecular systems involved in aging, cell proliferation, tissue renewal, oxidative stress, inflammation, immune competence, and carcinogenesis [28,29].

Several potent telomerase activators have been brought to market in recent years, based on their proposed action on telomeres in vitro [3,30]. Astragalus, due to its potent telomerase-activating compounds, has shown benefits in vitro and in animal experiments [31,32,33], and early human trials have produced encouraging results [34]. We recently reported the benefits of an Astragalus extract containing astragaloside IV and cycloastragenol, a potent telomerase activator, in an open prospective preliminary study on telomere size and cardiovascular impact in healthy volunteers [35]. Encouraged by these results, we conducted a randomized, double-blind, controlled trial over six months to compare the effect of this Astragalus-based nutritional supplement versus a placebo on TL in 40 healthy volunteers. The purpose of this study is, then, to validate that a natural astragalus-based nutritional supplement lengthens telomeres in a middle-aged population thanks a randomized, double-blind, placebo-controlled study.

## 2. Materials and Methods

### 2.1. Population Studied

We conducted a randomized, double-blind, placebo-controlled, parallel-group trial. Subjects were randomized to receive either a placebo or the active Astragalus-based supplement (ASTCOQ02) using a random number table. The participants, investigators, coordinators, and data analysts involved in the study were blinded to the identity of bottles A and B until the end of the study. Forty volunteer subjects with a mean age of 56.1 ± 6.0 years were included. There were 24 women (mean age: 54.8 ± 5.7 years) and 16 men (mean age: 57.3 ± 5.9 years), all ambulatory, with no significant medical history. Twenty subjects were randomized to group A to receive ASTCOQ02, and 20 subjects were randomized to group B to receive the placebo. Group A included 8 men and 12 women, with a mean age of 55.8 ± 6.5 years. Group B included 8 men and 12 women, with a mean age of 56.4 ± 5.6 years (NS).

The inclusion criteria were healthy volunteers without any of the exclusion criteria, with stable physical activity, and able to sign their informed consent. The exclusion criteria included non-menopausal women, subjects under 40 or over 70 years old, subjects with active cancer or a prior history of breast cancer, subjects with severe infectious disease, chronic disease (metabolic, cardiovascular, neurological), or autoimmune disease, or subjects receiving treatment with statins (or red yeast rice), danazol, rapamycin, oral antidiabetic drugs, corticoids, or non-steroidal anti-inflammatory drugs.

The two groups were similar at baseline in terms of the percentage of active and retired subjects, medical history, body mass index, weekly physical activity (4.4 ± 2.7 h/week in group A versus 3.8 ± 2.4 h/week in group B, *p* = 0.1), routine biochemical test values, immunological status, medical drugs or metabolic agents (with respect to study contraindications), and nutritional supplementation.

### 2.2. Study Design

The study was conducted at the Institute of Medicine and Physiology of Longevity (Institute de Jaeger) in Paris, France. The trial lasted six months, with subjects having five visits: preselection, day 0 (baseline), and visits at 1 month, 3 months, and 6 months (final visit).

Among the forty volunteers recruited, half of them received a 6-month course of ASTCOQ02, while the others received a 6-month course of placebo capsules. The placebo capsules contained excipients only. All 40 subjects completed the study.

All subjects signed their informed consent before inclusion in the protocol. All the volunteers were members of our medical and paramedical staff. None of the subjects had active cardiovascular disease at the time of inclusion. All changes in physical activity or drug intake during the study period (mean physical activity for the 40 volunteers: 4 h per week, stable for 3 years) were recorded, along with occurrences of any negative side effects.

The study adhered to the tenets of the Declaration of Helsinki. The French National Drugs Agency (ANSM) approved the study, and written informed consent was obtained. The National Advisory Committee on Information Processing in Material Research in the Field of Health (CCTIRS) and the National Commission on Informatics and Liberty (CNIL) also approved the trial.

The telomerase activator complex tested was ASTCOQ02, a blend of Astragalus extracts (including astragaloside IV and cycloastragenol), olive fruit extract (including hydroxytyrosol), zinc oxide, and grape seed extract. The dosage form was an oral capsule, already used in the clinical pre-study [35], taken twice a day (one in the morning and one in the evening) for 6 months. The daily dose delivered by the two capsules included astragalus extract: 250 mg; astragaloside IV: 40 mg; cycloastragenol: 25 mg; zinc oxide: 14.46 mg (144.6% DVR*); grape seed extract: 160 mg; olive fruit extract: 140 mg; and hydroxytyrosol and its derivatives: 28 mg.

The product, marketed as a food supplement, has no known toxicity. The content and appearance of the ASTCOQ02 and placebo capsules were indistinguishable in shape, size, weight, and packaging and followed the same twice-daily regimen.

### 2.3. Blood Samples

Venous blood samples were collected four times during the study (at day 0 (baseline) and then at 1, 3, and 6 months) and sent to the lab for analysis.

### 2.4. Measurement of TL

Telomere length was measured using the quantitative fluorescence in situ hybridization (Q-FISH) technique (Life Length, Madrid, Spain) as described elsewhere [35,36]. The Q-FISH methodology characterizes restriction fragments bearing TTAGGG repeats. It involves digesting the telomeric DNA with one or more restriction enzymes and then separating the restriction fragments according to their size using gel electrophoresis. The fragments are then transferred to a nitrocellulose or nylon membrane, which will bear a replica of the position of the fragments in the gel. The DNA is then denatured, arranged, and hybridized with a fluorescent probe, and the position of the fragments is revealed by radiography, showing one or more black bands. The size of the terminal fragments is estimated by comparing the distances they traveled in the gel against the distance covered by fragments of known length. This high-throughput Q-FISH technique measures the median and average telomere length, percentage of short telomeres, percentile of the shortest telomeres, and global distribution profile of the telomeres. It has the advantage of determining the length of any telomere, regardless of its size, but the disadvantage of being relatively long to run. the Results are expressed in kilobase pairs (kbp). All measurements were performed in quintuplicate. Short telomeres were defined as the percentage of telomeres with a length below 3 kbp (<3 kbp).

### 2.5. Clinical Laboratory Assays

During the visits at baseline and at the end of visits at 1, 3, and 6 months after the initiation of the test products (ASTCOQ02 or placebo), vitals were checked and blood was drawn from each subject. Each visit included a series of assays: a hematology panel (RBC, hemoglobin, hematocrit, complete blood count, white blood cell count, differential leukocytes, platelets), a metabolic panel (glucose, glycated hemoglobin, urea nitrogen, creatinine, estimated glomerular filtration rate, sodium, potassium, blood uric acid, bilirubin, alkaline phosphatase, aspartate aminotransferase, alanine aminotransferase), a lipid panel (total cholesterol, HDL cholesterol, triglycerides, and LDL cholesterol), inflammatory markers (C-reactive protein, homocysteine, TNF-alpha), plasma antioxidant potency (PAP) (total PAP in mmol/L—normal range between 1.35 and 1.65) (Dr. C. Garrel, Biochemistry Laboratory, Grenoble, France), and immune cells, including immune-senescence biomarkers (lymphocytes, T lymphocytes, CD4/CD8 ratio).

### 2.6. Statistical Analysis

The mean values and standard deviations of the overall population of 40 healthy volunteers and 2 independent groups (men and women) were computed as a function of clinical and biological parameters and telomere size. The averages of the different groups were computed using a parametric Student’s *t*-test, with a level of significance set at *p* < 0.05.

The baseline data were analyzed for cross-sectional age effects. Student’s *t*-tests were used to compare the means, and the F-distribution was used to determine the significance of the linear regression models against subject age. Except where indicated, two-tailed paired *t*-tests were performed at each time point for comparisons against the baseline values. For the percentage of short telomeres analyzed via high-throughput Q-FISH, a chi-squared test was used to analyze individual differences between baseline and post-product data.

All statistical analyses were performed using IBM SPSS Statistics 24.0 (IBM, North Castle, NY, USA). Graphs were plotted using Excel 365 for Windows (Microsoft Corp., Redmond, Washington, DC, USA). Statistically significant differences were declared at *p* < 0.05.

## 3. Results

Although the difference was not significant, the active-ingredient group decreased its physical activity during the study (from 4.4 ± 2.7 h/week at baseline to 3.3 ± 1.7 h/week at 6 months, *p* = 0.1), while the placebo group maintained stable physical activity over time (from 3.8 ± 2.4 h/week at baseline to 3.7 ± 2.4 h/week at M6).

### 3.1. Compliance and Tolerance

All 40 subjects completed the study, with no adverse side effects reported at 6 months. No subjects discontinued the treatment or placebo due to adverse events. There were no changes in weight, blood pressure, or heart rate. Biological and immunological assessments did not reveal any alterations. Medium-sensitive CRP remained unchanged during the study (mean CRP for group A was 1.08 mg/L at baseline and 1.15 mg/L at 6 months versus 1.09 mg/L at baseline and 1.02 mg/L at 6 months for the placebo group: NS). The T0, M1, M3, and M6 electrocardiograms showed no abnormalities in electrocardiogram trace and PQ space in an independent analysis by a single-blind cardiologist.

### 3.2. Changes in Telomere Length

#### 3.2.1. Median Telomere Length

At baseline, there were no significant between-group differences in median TL, although telomere size was higher in the placebo group (10.240 ± 0.746 kbp versus 9.805 ± 0.823 kbp in the ASTCOQ02 group, *p* = 0.17; Figure 1). In the placebo group, the median telomere length either decreased or remained stable at 1, 3, and 6 months compared to baseline (Figure 1). The median telomere length decreased by 93 kbp at 1 month (10.147 ± 0.308 kbp–NS), then remained stable at 3 months (10.353 ± 0.464 kbp–NS) and 6 months (10.437 ± 0.823 kbp–NS).

Conversely, in the ASTCOQ02 group, the median telomere length increased significantly compared to baseline (9.805 ± 0.823 kbp), starting at 1 month (+271 kbp longer, i.e., 10.076 ± 0.876 kbp, *p* = 0.3), then increasing at 3 months (+472 kbp longer, i.e., 10.277 ± 0.742 kbp, *p* = 0.008) to become significantly higher at 6 months (+695 kbp longer, i.e., 10.507 ± 0.679 kbp, *p* = 0.001; Figure 1).

The time effect was significantly different in the ASTCOQ02 group compared to placebo. ASTCOQ02 showed beneficial effects on median telomere length compared to the placebo, with regular size increases from the first month through 3 months and then 6 months (Figure 1).

#### 3.2.2. Short Telomere Length

At baseline, there were no significant between-group differences in median telomere length, although the mean size of short telomeres was higher in the placebo group (6.312 ± 0.982 kbp versus 5.773 ± 1.017 kbp; *p* = 0.10; Figure 2). In the placebo group, the mean short telomere length remained stable at 1, 3, and 6 months compared to baseline (Figure 2). The median short telomere length decreased slightly at 1 month, then remained stable at 3 months and 6 months (6.626 ± 0.793 kbp—*p* = 0.54 versus baseline; Figure 2).

Conversely, in the ASTCOQ02 group, the median short telomere length increased significantly compared to baseline, starting at 1 month (6.017 ± 0.974 kbp versus 5.773 ± 1.017 kbp at baseline, *p* = 0.44) and becoming significant at 3 months (6.423 ± 0.780 kbp, *p* = 0.029) and at 6 months (6.594 ± 0.648 kbp, *p* = 0.006; Figure 2). ASTCOQ02 showed beneficial effects on short telomere length compared to the placebo, with regular size increases from 1 month becoming significant at 3 and 6 months (Figure 2).

#### 3.2.3. Percentage of Short Telomeres

At baseline, there was no significant between-group difference in the percentage of short telomeres (<3 kbp) (5.725 ± 2.549% for the placebo versus 6.645 ± 3.680% for the ASTCOQ02 group, *p* = 0.06). The percentage of short telomeres tended to decrease in the placebo group (*p* = 0.16), but significantly decreased in the ASTCOQ02 group at 6 months (6.645 ± 3.680% at baseline versus 4.870 ± 1.468% at 6 months; *p* = 0.04; Figure 3).

## 4. Discussion

Recent reports have already shown promising results for several molecules belonging to the pharmaceutical class of telomerase activators [3,30]. Astragalus, a plant used in traditional Chinese medicine, is one of the most potent products in this pharmaceutical class. Astragalus and one of its derivatives (astragalosides) appear to be metabolized to cycloastragenol (CA), a telomerase activator [37,38]. Some products from Astragalus have shown benefits [31,32,33]. TA-65, an astragaloside IV, significantly increases telomerase activity 1.3 to 3.3-fold relative to controls in human T-cell cultures [39].

Our telomerase activator complex also contains hydroxytyrosol, which is known to inhibit oxidative stress and inflammation by enhancing the nuclear factor erythroid-2-related factor/heme-oxygenase 1 (Nrf2/HO-1) signaling pathway and inhibiting the mitogen-activated protein kinase/nuclear factor-kappa B (MAPK/NF-κB) signaling pathway [40,41]. So hydroxytyrosol can also have an influence on the final result. Another study involving a placebo containing olive fruit extract to demonstrate the effect of Astragalus extract could be interesting.

The randomized, double-blind, placebo-controlled study showed that a natural Astragalus-based nutritional supplement lengthens telomeres in a middle-aged population. Our previous report had already highlighted a significant increase in short telomere length between baseline and M6 in all 10 subjects included [35], prompting us to proceed with this randomized, double-blind, placebo-controlled study.

Indeed, the present longitudinal study confirmed a decline or a non-significant increase in both median and short telomere lengths in the placebo group, whereas the ASTCOQ02 group had a net increase in median telomere length of 271 kbp at 1 month, 472 kbp at 3 months, and 696 kbp at 6 months (*p* = 0.01) and a net increase in average short telomere length of 244 kbp at 1 month, 650 kbp at 3 months, and 810 kbp at 6 months (*p* = 0.004).

No subjects discontinued the treatment or placebo due to adverse events. This is in accordance with a preliminary study having reported good tolerance of the telomerase activator ASTCOQ02 in healthy volunteers [35]. Then, here, we further confirm that ASTCOQ02 administration did not lead to any product-related toxicities based on time-course assessments of key biochemical markers of liver, kidney, and metabolic function, as well as electrocardiogram assessments.

These highly significant results, obtained in this longitudinal investigation protocol, are of great interest given that prior cross-sectional studies have shown that telomere length, a marker of cellular aging, decreases with age. Indeed, humans slowly lose telomeric DNA at an average rate of 50 to 200 base pairs per year [42,43]. However, while telomere length declines with age, telomere size remains stable or can even transiently increase slightly over short periods of time [16]. In fact, the telomere point measurement has no intrinsic value per se. It is the measure of variation in telomere size, one way or the other, that essentially informs future quality of longevity.

Several studies have clearly demonstrated that telomere length correlates with senescence, including in endothelial cells [44], but also with numerous aging-related diseases, including cardiovascular atherosclerotic and inflammatory diseases, cancer, infections, stroke, mental pathologies, osteoporosis, and dementia [21,22,23,25,29,45,46,47,48].

Telomere size inversely correlates with obesity [49,50], smoking [10,51], stress and depression [52], type 2 diabetes [53], and high blood pressure [54]. However, diet and physical activity positively impact telomere strength and aging [11,55]. The two groups studied here were similar at baseline in terms of medical history, body mass index, weekly physical activity (4.4 ± 2.7 h/week in group A versus 3.8 ± 2.4 h/week in group B; *p* = 0.1), routine biochemical test values, immunological status, medical drugs or metabolic agents, and nutritional supplementation. Furthermore, the fact that physical activity decreased over the course of the study (by 1.1 h/week) in the ASTCOQ02 group but not in the placebo group reinforces the results reported here.

The link between telomere shortening and cardiovascular pathology has been the focus of extensive research. The roles of chronic inflammation, oxidative stress, and telomere shortening in the onset of atherosclerosis remain debatable [49,56]. In many cases, coronary diseases emerge in settings marked by atherothrombosis and cellular senescence, which have been the focus of attention for many years [57,58,59]. It has been suggested that the length of telomeres within vascular cells plays a key role in the development of coronary pathologies by establishing a particular senescence phenotype in smooth muscle and endothelial cells [60,61]. Brouilette et al. (2008) led a case–control study of 104 subjects (45 with a family history of coronary pathology and 59 control subjects) and found an association between family history of coronary disease and telomere shortening [62]. These results suggest that the presence of short telomeres is one of the primary abnormalities of atherosclerotic coronary disease and a good predictor of risk for cardiovascular events.

More recently, Goglin et al. (2016) showed that telomere size was closely related to cardiovascular mortality independently of common cardiovascular risk factors [63]. Depinho et al. (2011) reported that the activation of telomerases improved tissue degeneration in elderly mice [64].

Here, we further confirmed results previously obtained in animal models that show that telomere shortening prevention therapy can improve cardiovascular disease and longevity. However, there are currently few data on possible therapeutic strategies that can protect telomere length or drive telomere lengthening, particularly in the context of cardiovascular disease. Physical exercise appears to positively influence telomere length [54,65,66,67].

The WOSCOPS study showed that statins could have a beneficial effect on telomere length, as in addition to their lipid-lowering effect, they help preserve vessel wall integrity in at-risk patients by decreasing the senescence of endothelial cells [11,55]. Oral antidiabetic drugs such as pioglitazone, which belongs to another pharmacological class, appear to have a beneficial effect on telomere biology by increasing telomerase activity [44,45]. However, no clinical studies have been reported to date. At present, no other therapeutic classes (fibrates, NSAIDs, aspirin, corticosteroids, beta-blockers, insulin, ACE inhibitors, angiotensin II receptor antagonists, etc.) seem to have any effect on telomere biology. Our study thus opens a novel and effective therapeutic pathway to control telomere length in aging and/or support the prevention of cardiovascular-related diseases. This work has been confirmed in many animal models. However, there are currently few data on possible therapeutic strategies that can be used to protect telomere length or drive elongation, particularly in the context of cardiovascular disease [65,66,67,68]. Physical exercise appears to positively influence telomere length [69].

In agreement with previous in vitro studies, this randomized study showed that ASTCOQ02 can increase telomerase length, could prevent cell senescence, and may lead to longer life expectancy and healthy aging.

Limitations of this study: The design of the present study did not allow us to compare the nutrient effect of this natural Astragalus-based nutritional supplement on Telomere length in this middle-aged population. Additionally, according to a prior study, women of the same age exhibit a lower within-individual leukocyte telomere length shortening rate than men, but here, we did not specifically study the gender effect [16]. A further randomized, double-blind, placebo-controlled study is therefore warranted.

## 5. Conclusions

This randomized, double-blind, placebo-controlled study confirmed that ASTCOQ02 lengthens both median and short telomeres by increasing telomerase activity and reduces the percentage of short telomeres (<3 kbp). In addition, our results further confirm our previous open prospective preliminary study that found zero toxicity associated with the intake of ASTCOQ02. This randomized, double-blind, placebo-controlled trial confirmed that ASTCOQ02 can lengthen telomeres in a statistically and possibly clinically significant manner. ASTCOQ02 warrants further research to investigate its pro-health benefits for healthy aging and longer life expectancy.

## Figures and Tables

**Figure 1 nutrients-16-02963-f001:**
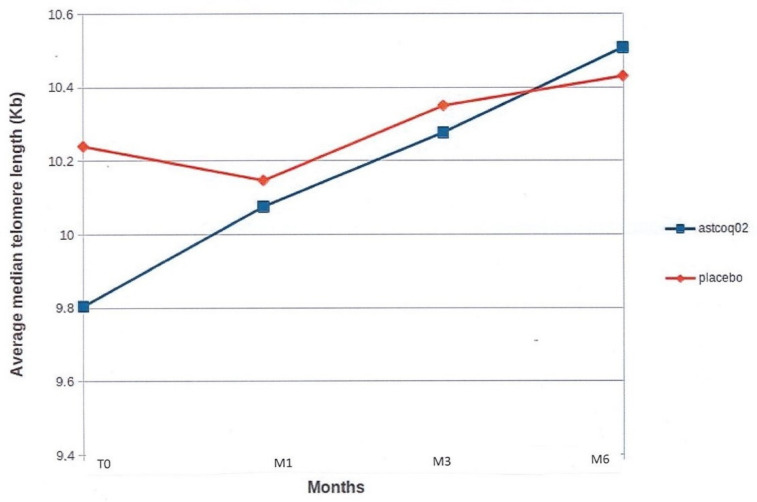
Median telomere length. This figure shows the average median telomere length in kilobases (Kb) over time for two groups: one treated with ‘ASTCOQ02′ and one with a ‘placebo’. The ‘ASTCOQ02′ group starts at 9.8 Kb at T0 and increases to 10.4 Kb at M6 (*p* = 0.001). The ‘placebo’ group starts at 10.2 Kb at T0, with minor fluctuations, ending at 10.3 Kb at M6 (NS).

**Figure 2 nutrients-16-02963-f002:**
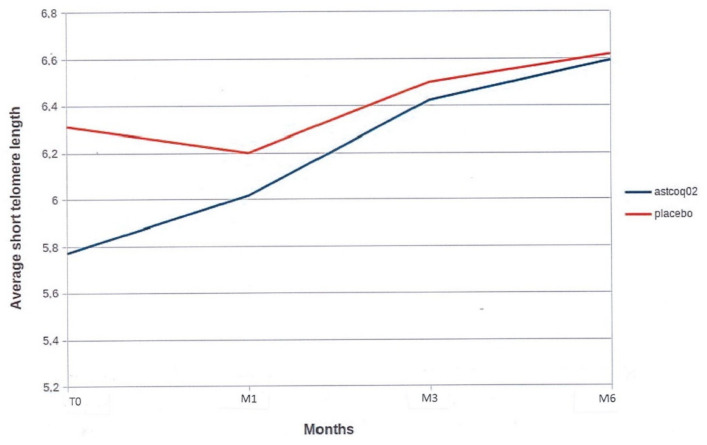
Average short telomere length. This figure illustrates the average short telomere length over time for two groups: one treated with ‘ASTCOQ02′ and one with a ‘placebo’. The y-axis represents the average short telomere length in kilobases (Kb), ranging from 5.2 to 628. The x-axis shows the time points T0, M1, M3, and M6. The blue line represents the ‘ASTCOQ02′ group, which starts at 5.8 Kb at T0 and increases to 6.6 Kb at M6 (*p* = 0.006). The red line represents the ‘placebo’ group, which starts at 6.4 Kb at T0, with minor fluctuations, ending at 6.6 Kb at M6 (NS).

**Figure 3 nutrients-16-02963-f003:**
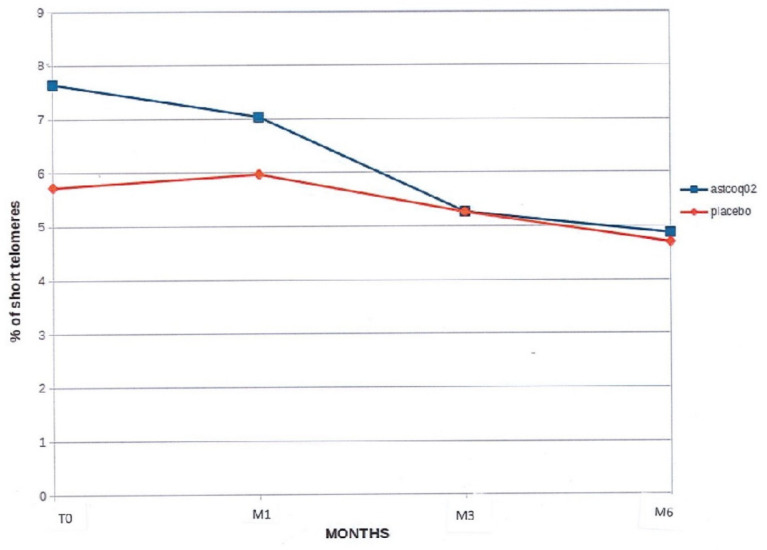
Percentage of short telomeres. This figure shows the percentage of short telomeres over time for two groups: one treated with ‘ASTCOQ02′ and one with a ‘placebo’. The y-axis represents the percentage of short telomeres, ranging from 0% to 9%. The x-axis shows the time points T0, M1, M3, and M6. The blue line represents the ‘ASTCOQ02′ group, which starts at 8% at T0 and decreases to approximately 5.5% at M6 (*p* = 0.04). The red line represents the ‘placebo’ group, which starts at 6% at T0 and remains relatively stable, ending at about 5.5% at M6 (NS).

## Data Availability

The datasets generated and/or analyzed during the current study are available from the corresponding author on reasonable request.

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
