# Peer review of "A Natural Astragalus-Based Nutritional Supplement Lengthens Telomeres in a Middle-Aged Population: A Randomized, Double-Blind, Placebo-Controlled Study"

_nutrients, 2024, doi:10.3390/nu16172963_

Round 1

Reviewer 1 Report

Comments and Suggestions for Authors

The manuscript - for me and I hope for the medical world as well - is interesting and brings results that give hope for a healthy and longer life.

To be corrected/completed in order to improve the manuscript:

- Latin names are written in italics (e.g. Astragalus)

- to specify the species of Astragalus or Astragalus sp. (if it is secret!) - everywhere in the manuscript.

- In the Abstract, specify that "Subjects taking the Astragalus-based supplement exhibited significantly longer median TL (P=0.01) and short TL (P=0.004), along with a lower percentage of short telomeres, over the six-month period, while the placebo group showed no change in TL." What did you compare it to? Do you have an initial measurement, before administration? Or did the supplement lead to an increase in telomere length?

- to be corrected (line 364): ASTCOQ02 can increase telomerase length, could prevent cell senescence, and may lead...

Author Response

Paris, 24th of August, 2024.

Dear Reviewer,

I wanted to thank you first of all for the time and attention you have given to our work.

I have answered you as best I could following each of your comments.

I obviously remain at your disposal for any additional questions.

Comments and Suggestions for Authors

  1. The effect of Mediterrenial diet on the telomere preservation has been studied (as review, recently i. a. by Baliou et al. in Nutrients). Hydroxytyrosol i san important component of this diet; its antiaging action has been demonstrated and its contribution to the effect studied can be considered. To demonstrate the effect of the Astragalus extract, a study inolving placebo containing the olive-fruit extract would be desirable.
  2. In this preliminary work, we sought to test a molecular complex whose components are known to be active in telomere lengthening. At this stage, we did not wish to study the value of each component of the complex.

But of course, that could be the subject of another work.

We have also included in the discussion the work of Baliou and all, in Nutrients, which is entirely relevant and enriching.

  1. Title: The meaning of „Natural”  is not fully obvious; the preparation used contained Astragalus extract but it was a preparation containing natural components and not a natural substance per se.
  2. You are absolutely right about the title in substance, but we have kept it the same so as not to lengthen it and overload it. All explanations were then given in the chapter "materials and methods".

  1. Material and methods: What was the source of ASTCOQ02?
  2. The complex was manufactured by a renowned French manufacturer (Fenioux laboratories).

  1. Lines 153-159: Please do not punctuate ingredients contained in the preparations to avoid impression that they are independent additives.
  2. I made the change.

  1. Line 196: Is serum TNF-alpha a marker of senescence or of inflammation?
  2. TNF alpha is used as an inflammation marker. I change the text to make it clearer.

  1. Figures: please indicate statistically significant differences
  2. Done.

I hope I have answered your questions and suggestions and I thank you again for your kind attention.

Kind regards

Dr Christophe de JAEGER

Open Review

Quality of English Language

( ) I am not qualified to assess the quality of English in this paper.
( ) The English is very difficult to understand/incomprehensible.
( ) Extensive editing of English language required.
( ) Moderate editing of English language required.
( ) Minor editing of English language required.
(x) English language fine. No issues detected.

Yes

Can be improved

Must be improved

Not applicable

Does the introduction provide sufficient background and include all relevant references?

( )

(x)

( )

( )

Is the research design appropriate?

( )

( )

(x)

( )

Are the methods adequately described?

( )

( )

(x)

( )

Are the results clearly presented?

( )

(x)

( )

( )

Are the conclusions supported by the results?

( )

( )

(x)

( )

Comments and Suggestions for Authors

The results of studies presented in an article entitled: A Natural Astragalus-Based Nutritional Supplement Lengthens Telomeres in a Middle-Aged Population: A Randomized, Double-Blind, Placebo-Controlled Study have taken into consideration the influence of adaptogen Astragalus on the telomere length. The influence of natural compounds on health and genetic information stability is well known as for the plethora of antioxidants. However, the topic of the manuscript can be interesting, especially in the case of some neurodegenerative disorders and cancers. The experimental and statistical methods have been well described (Measurement of telomers length, clinical assays, and statistics). However, in the article, I did not find the results derived from the reference group, random patients who did not receive diet supplements and placebo. Moreover, the authors did not provide the reference number of the bioethics commission agreements number. The sentence “The study adhered to the tenets of the Declaration of Helsinki. The French National Drugs Agency (ANSM) approved the study, and written informed consents were obtained. The National Advisory Committee on Information Processing in Material Research in the Field of Health (CCTIRS) and the National Commission on Informatics and Liberty (CNIL) also approved the trial.” is not sufficient and did not provide the critical information. Moreover, the authors did not provide any phytochemical analysis of the discussed adaptogen the description: Astragalus extract: 250 mg is without the necessary information, especially in the age of high-resolution LC-MS/MS.

The article is well-written and readable moreover the references have been correctly selected and properly inserted in the manuscript text.

In conclusion due to the critical remarks presented above I cannot recommend this article for publication in Nutrients journal.

Submission Date

30 July 2024

Date of this review

14 Aug 2024 13:10:16

Reviewer 2 Report

Comments and Suggestions for Authors

I would like to congratulate authors on this interesting study, demonstrating that a suplement containing a natural product may activate telomerase and increase telomere length in humans during a 6-month experiment.

The group studied was not too numerous; therefoe, the results should be treated as rather preliminary but, nevertheless, are very interesting. What should be discussed is the presence of hydroxytyrosol in the preparation. The effect of Mediterrenial diet on the telomere preservation has been studied (as review, recently i. a. by Baliou et al. in Nutrients). Hydroxytyrosol i san important component of this diet; its antiaging action has been demonstrated and its contribution to the effect studied can be considered. To demonstrate the effect of the Astragalus extract, a study inolving placebo containing the olive-fruit extract would be desirable.

Otherwise the study design is correct. The results are properly presented and discussed.

Other remarks:

Title: The meaning of „Natural”  is not fully obvious; the preparation used contained Astragalus extract but it was a preparation containing natural components and not a natural substance per se.

Material and methods: What was the source of ASTCOQ02?

Lines 153-159: Please do not punctuate ingredients contained in the preparations to avoid impression that they are independent additives.

Line 196: Is serum TNF-alpha a marker of senescence or of inflammation?

Figures: please indicate statistically significant differences

Please format References according to the requirements of the journal.

Author Response

(The authors gave the same response as above.)

Reviewer 3 Report

Comments and Suggestions for Authors

The results of studies presented in an article entitled: A Natural Astragalus-Based Nutritional Supplement Lengthens Telomeres in a Middle-Aged Population: A Randomized, Double-Blind, Placebo-Controlled Study have taken into consideration the influence of adaptogen Astragalus on the telomere length. The influence of natural compounds on health and genetic information stability is well known as for the plethora of antioxidants. However, the topic of the manuscript can be interesting, especially in the case of some neurodegenerative disorders and cancers. The experimental and statistical methods have been well described (Measurement of telomers length, clinical assays, and statistics). However, in the article, I did not find the results derived from the reference group, random patients who did not receive diet supplements and placebo. Moreover, the authors did not provide the reference number of the bioethics commission agreements number. The sentence “The study adhered to the tenets of the Declaration of Helsinki. The French National Drugs Agency (ANSM) approved the study, and written informed consents were obtained. The National Advisory Committee on Information Processing in Material Research in the Field of Health (CCTIRS) and the National Commission on Informatics and Liberty (CNIL) also approved the trial.” is not sufficient and did not provide the critical information. Moreover, the authors did not provide any phytochemical analysis of the discussed adaptogen the description: Astragalus extract: 250 mg is without the necessary information, especially in the age of high-resolution LC-MS/MS.

The article is well-written and readable moreover the references have been correctly selected and properly inserted in the manuscript text.

In conclusion due to the critical remarks presented above I cannot recommend this article for publication in Nutrients journal.

Author Response

(The authors gave the same response as above.)

Round 2

Reviewer 3 Report

Comments and Suggestions for Authors

Please add the number of the ethics commission agreement to the experimental part – Sine qua non - after it, in my opinion article can be accepted for publication.

Author Response

Paris, 28th of August, 2024.

Dear Reviewer,

I wanted to thank you first of all for the time and attention you have given to our work.

I obviously remain at your disposal for any additional questions.

Comments and Suggestions for Authors

  1. Please add the number of the ethics commission agreement to the experimental part – Sine qua non - after it, in my opinion article can be accepted for publication.

  1. The number of the ethics commission agreement is : 2019-A02675-52. I will integrate this number in the paper.

I hope I have answered your questions and suggestions and I thank you again for your kind attention.

Kind regards

Dr Christophe de JAEGER

Submission Date

30 July 2024

Date of this review

28 Aug 2024 11:03:42
